# High-throughput identification of synthetic riboswitches by barcode-free amplicon-sequencing in human cells

Benjamin Strobel [1,5], Maike Spöring [2,3,5], Holger Klein [4], Dragica Blazevic[1], Werner Rust[4], Sergi Sayols[4], Jörg S. Hartig[2,3,6] & Sebastian Kreuz [1,6]*

Synthetic riboswitches mediating ligand-dependent RNA cleavage or splicing-modulation represent elegant tools to control gene expression in various applications, including next-generation gene therapy. However, due to the limited understanding of context-dependent structure–function relationships, the identification of functional riboswitches requires large-scale-screening of aptamer-effector-domain designs, which is hampered by the lack of suitable cellular high-throughput methods. Here we describe a fast and broadly applicable method to functionally screen complex riboswitch libraries (~$1.8 \times 10^4$ constructs) by cDNA-amplicon-sequencing in transiently transfected and stimulated human cells. The self-barcoding nature of each construct enables quantification of differential mRNA levels without additional pre-selection or cDNA-manipulation steps. We apply this method to engineer tetracycline- and guanine-responsive ON- and OFF-switches based on hammerhead, hepatitis-delta-virus and Twister ribozymes as well as U1-snRNP polyadenylation-dependent RNA devices. In summary, our method enables fast and efficient high-throughput riboswitch identification, thereby overcoming a major hurdle in the development cascade for therapeutically applicable gene switches.

---

[1] Research Beyond Borders, Boehringer Ingelheim Pharma GmbH & Co. KG, Birkendorfer Str. 65, 88397 Biberach an der Riss, Germany. [2] Department of Chemistry, University of Konstanz, Universitätsstraße 10, 78464 Konstanz, Germany. [3] Konstanz Research School Chemical Biology (KoRS-CB), University of Konstanz, Universitätsstraße 10, 78464 Konstanz, Germany. [4] Computational Biology & Genomics, Boehringer Ingelheim Pharma GmbH & Co. KG, Birkendorfer Str. 65, 88397 Biberach an der Riss, Germany. [5]These authors contributed equally: Benjamin Strobel, Maike Spöring. [6]These authors jointly supervised this work: Jörg S. Hartig, Sebastian Kreuz. *email: sebastian.kreuz@boehringer-ingelheim.com

Artificial aptazyme riboswitches are fusions of a ligand-binding RNA aptamer and a self-cleaving ribozyme. Conditional cleavage is mediated by binding of a specific ligand by the aptamer domain and subsequent conformational changes in riboswitch architecture. Riboswitches can be encoded into DNA, e.g., in the 3′-untranslated region (UTR) upstream of a poly(A) signal of an expression construct to conditionally cleave the mRNA transcript[1]. Self-cleavage within the 3′-UTR can be exploited for the post-transcriptional control of gene expression by decreasing mRNA stability via conditional poly(A) tail cleavage. Utilization of riboswitches to control gene expression has been demonstrated in cells[2–4], *C. elegans*[5] and in a viral vector-mediated fashion in mice[6,7]. In addition to ribozyme-based switches, further synthetic mechanisms for gene expression control have been described in recent years including splicing control[8] and microRNA switches[9]. Due to their mRNA-intrinsic mode-of-action, independence of additionally expressed (potentially immunogenic) regulators, their small size and adaptability to various ligands, riboswitches represent an attractive technology for the development of gene expression control systems for viral vector gene therapy. Yet, to fully exploit their potential, optimized riboswitch designs enabling potent regulation in response to clinically applicable ligands are required.

To identify and optimize functional riboswitch designs, artificial riboswitch sequences are usually mutated at defined positions, e.g., within the linker sequence (also called the communication module) that connects the aptamer and effector domains. Riboswitch sequences are then screened for their activity, either by direct assessment of self-cleavage in in vitro transcription-based experiments (in the case of aptazymes) or by measuring the expression of riboswitch-regulated reporter proteins in bacteria, yeast or cell culture. Historically, these assessments have been conducted on the level of individual constructs, thereby limiting throughput to a maximum of a few hundred constructs, which can be tested by hand. More recently, higher-throughput methods have been described in yeast[10] and human cell culture[11]; however, these methods rely on time-, labor- and cost-intensive FACS-sorting or gel-electrophoretic pre-selection steps and sequence barcoding, respectively. Therefore, there is a high need for straightforward, time- and cost-effective screening methods to enable systematic assessment of different riboswitch design features in human cells, including new combinations of aptamers and ribozymes as well as communication module sequence, length, and symmetry.

Here we introduce a fast and generally applicable method for the identification of functional riboswitches in human cells, based on the detection of differences in mRNA levels by deep sequencing (Fig. 1). More specifically, following transfection of a plasmid-based riboswitch library in human HEK-293 cells and incubation in presence or absence of a ligand, RNA is extracted and reverse transcribed into cDNA. The primer binding site-flanked riboswitch sequence is PCR-amplified and analyzed by cDNA amplicon-seq. Because each riboswitch construct in a synthetic library is characterized by at least one single point mutation compared to every other sequence, each construct is self-barcoding, dismissing the need for additional cDNA manipulation steps. Differential mRNA levels (stimulated vs. unstimulated) are finally calculated and riboswitch sequences exhibiting desired expression changes are identified.

To establish this method and to demonstrate its ability to identify functional riboswitch constructs, we first screened libraries based on previously described tetracycline (Tet)-hammerhead (HHR), guanine (Gua)-hepatitis-delta-virus (HDV) and Gua-hammerhead ribozyme designs[4,12,13]. Using this approach, we recovered known and identified new Tet-responsive ON- as well as Gua-responsive OFF- and ON-switch sequences, respectively. In order to further validate the method's utility for the selection of functional constructs from de novo-designed libraries, we next

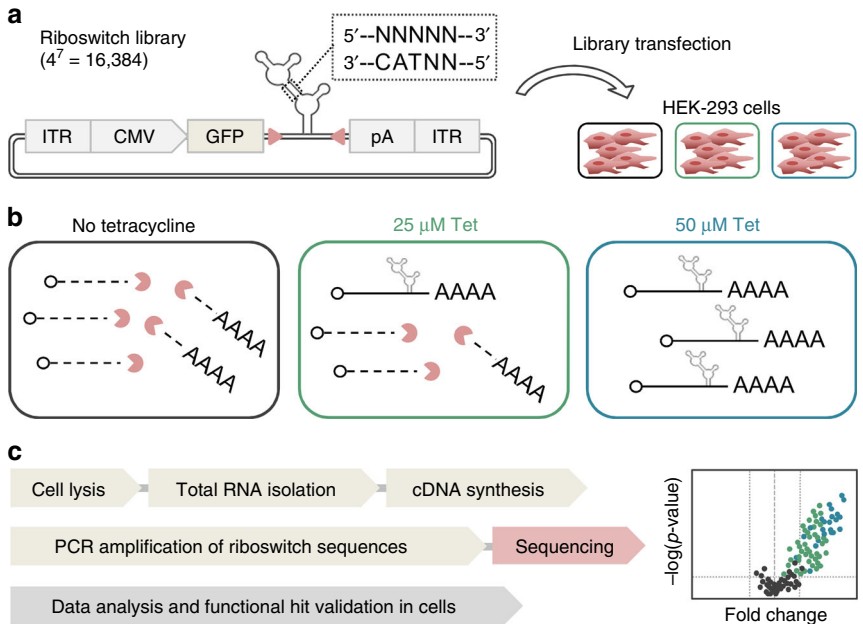

**Fig. 1 Principle and schematic workflow for the sequencing-based identification of functional riboswitches in human cells. a** HEK-293 cells are transfected with a plasmid library harboring PCR-binding site (red arrowheads)-flanked aptazyme variants with a randomized motif (here: tetracycline (Tet)-hammerhead design). **b** Functional ON-switches are characterized by aptazyme auto-cleavage and mRNA-degradation in the absence of Tet, whereas upon Tet addition, cleavage is inhibited, resulting in stable mRNA. **c** Following cell lysis, RNA purification and reverse transcription into cDNA, riboswitch sequences are PCR-amplified and applied to amplicon-seq analysis. Functional sequences are identified from differential expression analyses (stimulated vs. unstimulated). ITR, inverted terminal repeat; CMV, cytomegalovirus; GFP, green fluorescent protein; pA, polyadenylation signal.

generated a riboswitch library by connecting the tetracycline aptamer to the Twister ribozyme via a permutated communication module. Remarkably, screening of the Tet-Twister riboswitch library resulted in the identification of the first set of allosterically controllable Twister ribozymes for gene expression control in human cells. To demonstrate the broad applicability of our method beyond ribozyme-based designs, we finally describe the identification of functional constructs from a library based on a new mode of action, namely the ligand-dependent occupation of U1-snRNP binding sites, thereby conditionally controlling mRNA polyadenylation. Our amplicon-seq approach is characterized by a high sensitivity and a low false-positive discovery rate, thereby enabling the efficient and robust identification of RNA-based genetic switches. Finally, by developing a tailored computational analysis pipeline, comprehensive analyses for several quality measures, hit selection and structure–function characterization as well as graphical reporting thereof, are enabled.

## Results

### cDNA amplicon-seq identifies Tet-ON- and Gua-OFF-switches.

To establish our screening method, we made use of a previously reported Tet-hammerhead riboswitch design published by

Beilstein and colleagues (ref. [4] and Fig. 2a). This design is based on the fusion of the Tet aptamer[14] via its P1 stem to stem 1 of the full-length hammerhead N79 ribozyme[15]. Within the communication module connecting the aptamer and ribozyme, Beilstein and colleagues had altered up to five nucleotides on the 5′-strand and two on the 3′-strand based on mechanistic considerations to construct gene expression-inducing ON-switches (ref. [4] and Fig. 2a). In their experiments, ten out of seventeen tested mutant constructs demonstrated Tet-dependent functionality in HeLa cells. Building on these results, we constructed a highly extended library by fully randomizing the aforementioned seven nucleotides, resulting in a total of $4^7 = 16{,}384$ constructs (Fig. 2a and Supplementary Fig. 1a for sequence). Riboswitches were placed into the 3′-UTR of a CMV-eGFP reporter plasmid, flanked by $(CAAA)_3$ spacers and synthetic PCR primer binding sites. The plasmid library was quality-controlled by measuring the abundance of each individual library plasmid construct by DNA sequencing (Fig. 2b). Moreover, by using cDNA-amplicon-sequencing, we confirmed that construct abundance and library complexity were preserved after transfection into HEK-293 cells (Fig. 2c). As expected, only constructs with low initial library abundance displayed stronger fluctuations (defined as a coefficient of variation >10%) when comparing pre- and

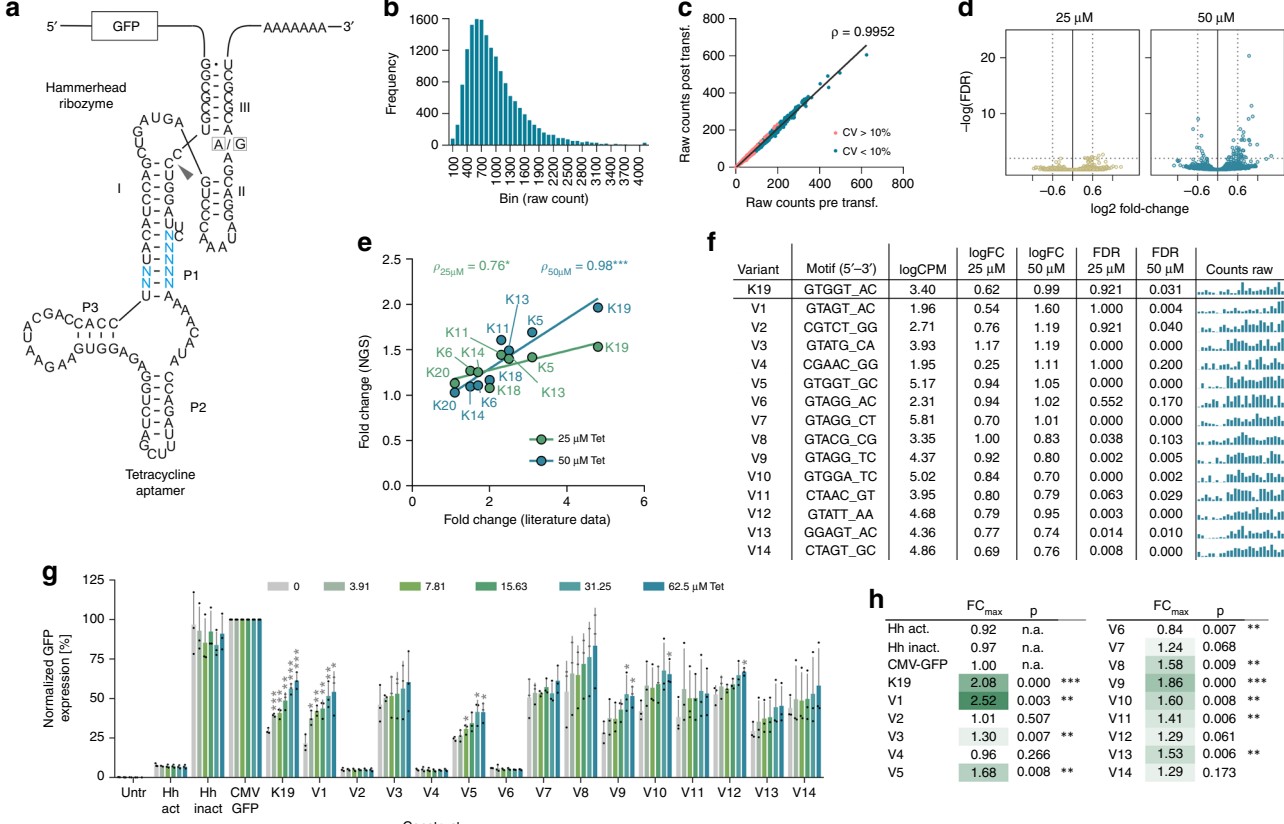

**Fig. 2 Identification of Tet-responsive hammerhead ribozymes by amplicon-seq. a** Secondary structure of the Tet-hammerhead ribozyme library design. Blue nucleotides indicate the randomized motif. The arrowhead indicates the cleavage site. A/G mutation at the indicated position renders the ribozyme inactive. **b** Tet riboswitch construct frequency analysis of the original library and **c** conservation of complexity following transfection in HEK-293 cells. CV, coefficient of variation. **d** Volcano plots displaying the significance (FDR) of changes in riboswitch abundance as a function of the change in expression (log₂ fold change). **e** Correlation of expression induction upon Tet for constructs and associated fold changes reported by Beilstein et al.[4] (X-axis) with fold changes measured in our screen (Y-axis). **f** Selected hits from the Tet riboswitch library screen, randomized sequence, mean counts per million (CPM), fold change (FC) and false-discovery rate (FDR) at 25 and 50 µM Tet, respectively. The bar plots show the raw counts under untreated, 25 µM and 50 µM Tet-stimulated conditions (three groups from left to right, n = 8 replicates each). **g** Functional hit validation in HEK-293 cells (n = 3 experiments, mean ± s.d.); Hh-act/inact: constitutively active/inactive ribozyme control. **h** Maximal fold changes and p-values measured upon stimulation vs. 0 µM Tet for all tested constructs. *p < 0.05, **p < 0.01, ***p < 0.001 (unpaired T-Test, two-tailed). Source data are provided as a Source Data file.

post-transfection, whereas a high library-wide correlation ($\rho =$ 0.9952) was observed (Fig. 2c).

To identify functional riboswitches, HEK-293 cells were transfected with the library and cultivated for 48 h, before tetracycline was added to the cells at increasing concentrations (25 and 50 μM). After 3 h of stimulation, cells were lysed and RNA was purified using silica column purification, including thorough DNA depletion. Following reverse transcription and non-saturating PCR-amplification of riboswitch sequences, PCR amplicons were purified, quality-controlled using the Fragment analyzer device and applied to sequencing library preparation. cDNA-amplicon-seq was conducted on an Illumina NextSeq 500 using 10 million single-end reads of 154 bp length per sample.

cDNA-amplicon-seq functional screening data were analyzed by first normalizing all constructs' cDNA amplicon counts to a control plasmid co-transfected with the library and then comparing the normalized counts of each riboswitch construct in presence vs. absence of ligand. Differential expression (i.e., fold-changes and Benjamini-Hochberg adjusted $p$-values/false-discovery rates (FDR)) was calculated, indicating enrichment of ON-switches with increasing Tet-dose (Fig. 2d) and rankings were generated based on the calculated expression changes at each Tet concentration. Dose-dependency was interpreted as an additional confidence factor.

Interestingly, among the top ranking hits based on the fold change upon stimulation with 50 μM Tet (i.e., on rank 17 out of 16,384), we found the "K19" variant, which was the most powerful Tet-hammerhead riboswitch identified by Beilstein et al. in their study (see top 100 in Supplementary Table 1)[4]. Moreover, a high Tet dose-dependent correlation between the originally reported expression changes and those measured in our screen was found for all previously reported functional constructs present in our library (Fig. 2e). Out of the hit list we selected the ten constructs showing the highest fold changes at 25 and 50 μM Tet each (FDR < 20%). After removal of three duplicate hits and three of four constructs with an AT-content >70% that was predicted to disturb overall riboswitch structure (construct V12 was kept due to the GTA-motif that seemed to be enriched among the selected hits), 14 constructs remained and were applied to functional validation in HEK-293 cells (Fig. 2f). Upon individual transfection and addition of increasing Tet doses, 10 of these constructs exhibited Tet dose-dependent induction of gene expression at varying potency (Fig. 2g), corresponding to a true positive rate of 71.4%. Expression was normalized to a conventional, riboswitch-free CMV-eGFP construct and additionally compared to K19 and constitutively active and inactive ribozyme variants. While K19 showed 2.08-fold induction across three experiments, the other constructs demonstrated maximal dynamic ranges of 1.39–2.52-fold (Fig. 2h). The three non-functional constructs displayed strongly reduced basal eGFP expression, indicative of constitutive ribozyme activity. Notably, two of these constructs had a screening FDR value > 0.1, suggesting that this cutoff is important during hit selection. In fact, if that cutoff would have been applied, the true positive rate would have further increased to 83.3%, as opposed to 58.8% (10/17 constructs) reported for manual testing. Notably, despite the identification of several riboswitch variants with different basal expression levels and dissimilar dynamic ranges, only one switch with a better performance, i.e. a higher dynamic range than K19, was identified in this screen. This finding demonstrates that the chosen design, but not the screening approach, in general, is the limiting factor for switch potency. Interestingly, the most potent construct (V1) differed from K19 only by a single nucleotide substitution. A high sequence similarity was also observed for many of the other functional sequences identified in this screen (Fig. 2f).

To demonstrate the method's ability to identify functional constructs, independent of the aptamer or ribozyme used, we next screened a library based on a Gua-responsive hepatitis-delta-virus (HDV) aptazyme design previously described by the Yokobayashi group[12] (Fig. 3a and Supplementary Fig. 1b for sequence). We fully randomized the previously described 6-bp motif within the communication module, resulting in a library diversity of 4096 constructs. Following QC (Fig. 3b), the library was screened in HEK-293 cells in absence or presence of 30, 100, and 300 μM guanine. Differential expression analysis indicated a guanine dose-dependent enrichment of OFF-switch sequences (Fig. 3c). Strikingly, the GuaM8HDV construct, which represented the most powerful switch reported in the original paper was also identified as the top hit in our screen (Fig. 3d). Moreover, the top 60 constructs in our screen (based on the log$_2$FC at 300 μM guanine, see Supplementary Table 2) also contained the five other previously identified switches that showed a fold change of >1.4 in the original publication. In contrast, the two weakest original variants were found on rank 3108 and 3888, respectively, out of 4096 constructs in our screen. Finally, 9 of the top 10 identified constructs (selection criteria: log$_2$FC <= −0.6, FDR <= 0.01, guanine dose-dependency) showed Gua-dependent downregulation of GFP expression in HEK-293 cells (Fig. 3e, f). Taken together, our data clearly demonstrate the ability of the amplicon-seq approach to recover known and identify new switch variants from complex construct libraries.

**cDNA-amplicon-seq identifies Gua-hammerhead ON-switches**. We next applied the amplicon-seq approach to a library based on the guanine aptamer fused to the hammerhead type-III ribozyme (Fig. 4a). In a previous study, we used a rational approach to construct Gua-dependent ON-switches using this design[13]. Now, we aimed to utilize amplicon-seq to explore a larger sequence space by randomizing the communication module at seven positions (Fig. 4a and Supplementary Fig. 1c for sequence). Following QC (Fig. 4b), the library was screened in HEK-293 cells in presence or absence of 30, 100, and 300 μM guanine.

Differential expression analysis showed a guanine dose-dependent increase for many sequences with a trend towards higher potency for ON- vs. OFF-switches (Fig. 4c). From the hit list ranked by logFC at 300 μM guanine (selection criteria: log$_2$FC >= 0.6, FDR <= 0.01, guanine dose-dependency), we selected the seven top hits for functional validation in HEK-293 cells (Fig. 4d). Notably, all seven constructs demonstrated potent and dose-dependent ON-switch activity (Fig. 4e, f). The most interesting switches in terms of fold change and dynamic range were GuaB4HHR (5.0–34.2% of the expression levels of a conventional, riboswitch-free construct, 6.9-fold change) and GuaB6HHR (19.6–84.1%, 4.3-fold change). Given that the previous rationally designed variants only showed up to 2-fold expression induction[13], these results nicely demonstrate the power of our screen to identify higher-potency switches from large sequence space.

**cDNA-amplicon-seq identifies Tet-Twister aptazymes**. The recently identified Twister ribozyme represents one out of currently 9 known self-cleaving ribozyme motifs[16] that can serve as an expression platform for novel aptazymes. We have recently shown that the Twister ribozyme is a versatile platform to construct one- and two-input genetic switches in *E. coli* and *S. cerevisiae*[17]. However, no allosterically controllable riboswitches that are functional in human cells have been described for this ribozyme so far. Therefore, we next aimed to identify Tet-responsive Twister riboswitches. We designed Tet-Twister ribozyme candidates by connecting the Tet aptamer via its P2 stem to the P1 stem of the Twister ribozyme (Fig. 5a). This design was chosen

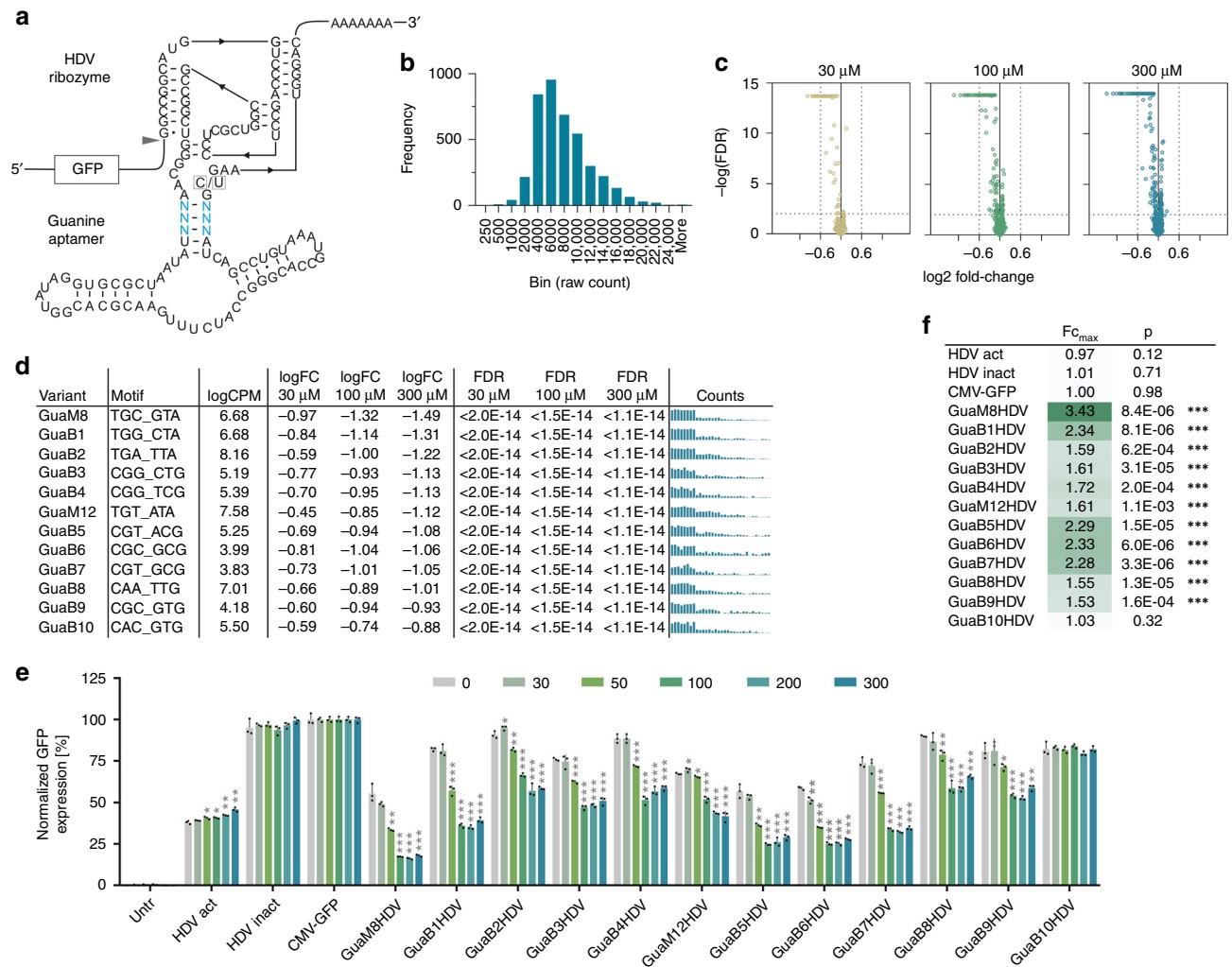

**Fig. 3 Identification of Gua-responsive HDV ribozymes by amplicon-seq. a** Secondary structure of the Gua-HDV ribozyme library design. Blue nucleotides indicate the randomized motif. The arrowhead indicates the cleavage site. C/U mutation at the indicated position renders the ribozyme inactive. **b** Guanine riboswitch construct frequency analysis of the original library. **c** Volcano plots displaying the significance (FDR) of changes in riboswitch abundance as a function of the change in expression (log₂ fold change). **d** Selected hits from the Gua-HDV riboswitch library screen, randomized sequence, mean counts per million (CPM), fold change (FC) and false-discovery rate (FDR) at 30, 100, and 300 μM guanine, respectively. The bar plots show the raw counts under untreated, 30, 100, and 300 μM μM Gua-stimulated conditions (four groups from left to right, $n = 8$ replicates each). **e** Functional hit validation in HEK-293 cells at increasing guanine doses [μM] ($n = 3$ replicates, mean ± s.d.); HDV-act/inact: constitutively active/inactive ribozyme control. **f** Maximal fold changes and p-values measured upon stimulation vs. 0 μM guanine for all tested constructs. *$p < 0.05$, **$p < 0.01$, ***$p < 0.001$ (unpaired T-Test, two-tailed). Source data are provided as a Source Data file.

based on our previous finding that the connection of the aptamer to the P1 site of the Twister ribozyme resulted in both, ON- and OFF-switches, which showed better performance than the designs based on connecting the aptamer to the P5 site[17]. Nine different sub-libraries were generated, where the length and stability ("CG" design, "noCG" design), symmetry and orientation of the communication module were varied (Fig. 5a and Supplementary Fig. 1d for sequence). The combination of three different lengths (by optional inclusion of CG pairs on either side of the communication module) with five (asymmetric linker, "2N3N" and "3N2N") or six randomized nucleotides (symmetric linker, "3N3N"), resulted in a final library complexity of 18,432 constructs. Library DNA sequencing was again applied to explore construct abundance. Due to the fact that the nine sub-libraries were mixed at equal ratios, the finding that 3N3N constructs were abundant at approximately four-fold lower levels than constructs in which only 5 nucleotides were randomized (2N3N and 3N2N), was expected (Fig. 5a, inset). The library was

screened in HEK-293 cells in absence or presence of 12.5, 25, and 50 μM Tet and construct functionality was analyzed by cDNA-amplicon-seq. Furthermore, in order to facilitate sequencing data processing, including quality control, data analysis and graphical output generation, a computational processing pipeline was built (available through GitHub, see 'Data availability" section for details). Among others, the quality measures included analyses of variant abundance on a single-sample level and its dependency on the GC-content of the randomized sequence motif. The analyses demonstrated that sample-to-sample fluctuation was low and that no GC-dependency (which could indicate potential biases in library synthesis) was observed (exemplarily shown in Supplementary Figs. 2 and 3).

The functional screening results clearly showed that our designs resulted in the selection of OFF-switches, whereas only a few motifs passed the significance threshold ($|\log_2 FC| >= 0.6$, FDR < 1%) for ON-switches (Fig. 5b and exemplary pipeline output in Supplementary Fig. 4). Interestingly, the observation that higher

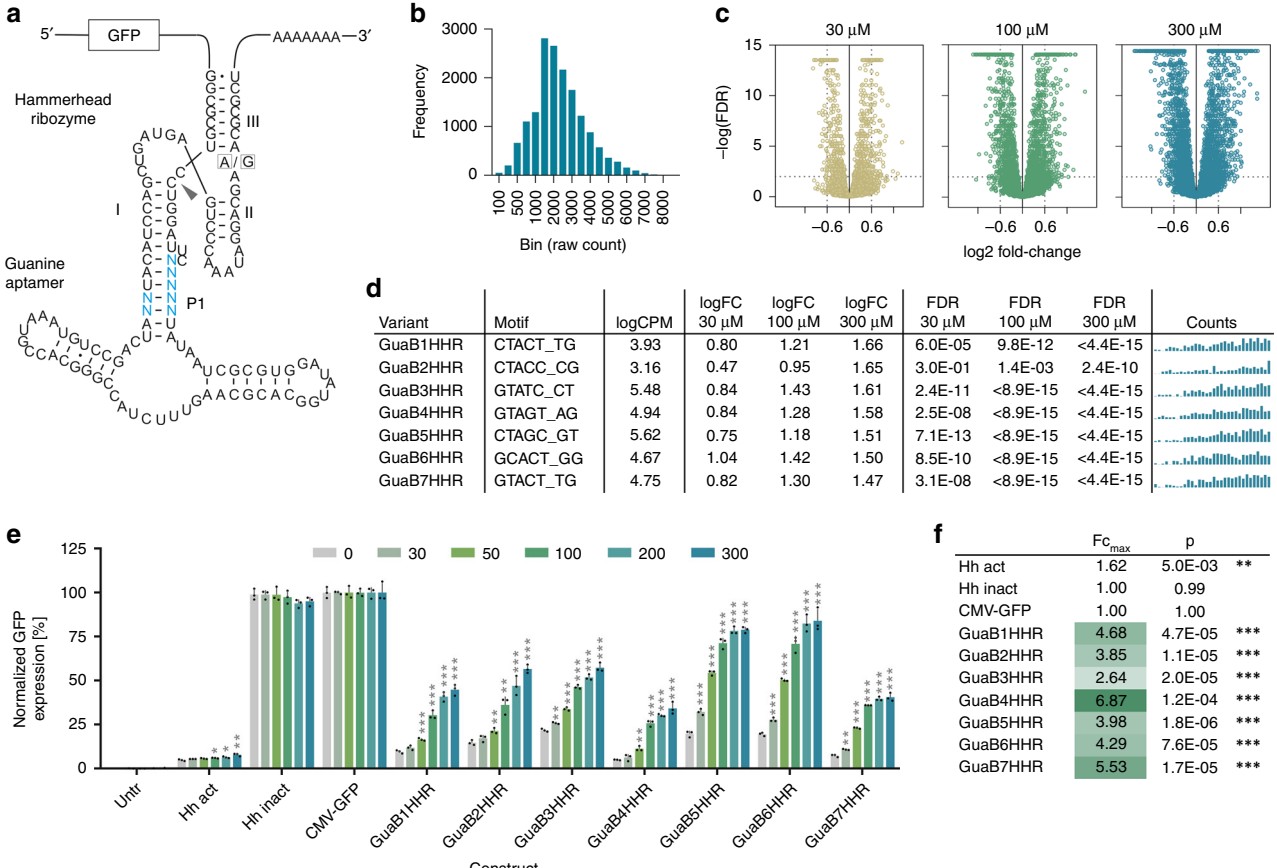

**Fig. 4 Identification of Gua-responsive hammerhead ribozymes by amplicon-seq. a** Secondary structure of the guanine hammerhead ribozyme library design. Blue nucleotides indicate the randomized motif. The arrowhead indicates the cleavage site. A/G mutation at the indicated position renders the ribozyme inactive. **b** Guanine riboswitch construct frequency analysis of the original library. **c** Volcano plots displaying the significance (FDR) of changes in riboswitch abundance as a function of the change in expression (log$_2$ fold change). **d** Selected hits from the Gua-HHR riboswitch library screen, randomized sequence, mean counts per million (CPM), fold change (FC) and false-discovery rate (FDR) at 30, 100, and 300 μM guanine, respectively. The bar plots show the raw counts under untreated, 30, 100, and 300 μM μM Gua-stimulated conditions (four groups from left to right, $n = 8$ replicates each). **e** Functional hit validation in HEK-293 cells at increasing guanine doses [μM] ($n = 3$ replicates, mean ± s.d.); Hh-act/inact: constitutively active/inactive ribozyme control. **f** Maximal fold changes and p-values measured upon stimulation vs. 0 μM guanine for all tested constructs. *$p < 0.05$, **$p < 0.01$, ***$p < 0.001$ (unpaired T-Test, two-tailed). Source data are provided as a Source Data file.

fold changes were observed for all 3N3N libraries compared to 2N3N and 3N2N designs further suggests that the symmetric communication module design leads to switches of higher potency than the asymmetric design. Moreover, the fraction of constructs displaying higher fold changes and/or higher significance was biggest in the "CG"-sub-libraries, suggesting that linker stabilization by the Twister-proximal CG pair has an overall beneficial effect. Conversely, the lower fold changes and less significant data obtained for the majority of the "noCG" variants indicate that the Tet aptamer-proximal CG pair is required for functionality. Variant similarity assessment among deregulated constructs based on Hamming distance and sequence motifs (exemplified in Supplementary Fig. 5) or similarity network analysis (Supplementary Fig. 6) resulted in groups of candidate constructs with related sequences.

Based on our screening results, we selected ten constructs (Fig. 5c and annotations in Fig. 5b) for functional validation in cells. Selected sequences were part of the top ten of each sub-library (Supplementary Table 3) in terms of fold change and had an FDR < 1% at more than one Tet concentration. They further showed dose-dependency and displayed maximal log$_2$FC values of greater than 1.0. Upon transfection of HeLa cells and stimulation with increasing doses of Tet, all ten candidate

constructs displayed OFF-switch activity, i.e. conditional and concentration-dependent downregulation of reporter gene expression (Fig. 5d). Notably, most of the "CG" design constructs (T5-T10) indeed showed higher fold changes than the constructs lacking the CG pair (Fig. 5e), as suggested by the screening results (Fig. 5b, c). Furthermore, as predicted, the top performing construct T9 was based on the symmetric 3N3N design. Finally, similarity network analysis of the CG_3N3N sub-library containing the T9 sequence (CCG_CAG) revealed several groups of related sequences (Fig. 5f and Supplementary Fig. 6), serving as a basis for future design strategies and assessment of structure–function relationships. The observation that all ten sequences selected for cellular characterization showed Tet-dose-dependent control of gene expression demonstrates that the screening approach together with the described selection criteria results in a low false-positive hit discovery rate. This notion is further supported by the finding that four randomly selected constructs that did not fulfill our selection criteria (i.e., they were found on ranks #500, #1000, #2000, and #3000 of a negative selection list based on a log2-fold change >−0.6 and an FDR > 0.01 at 300 μM) did not show switching behavior, as expected (Supplementary Fig. 7). In addition, the finding that switches with intermediate to low performance were identified in both of

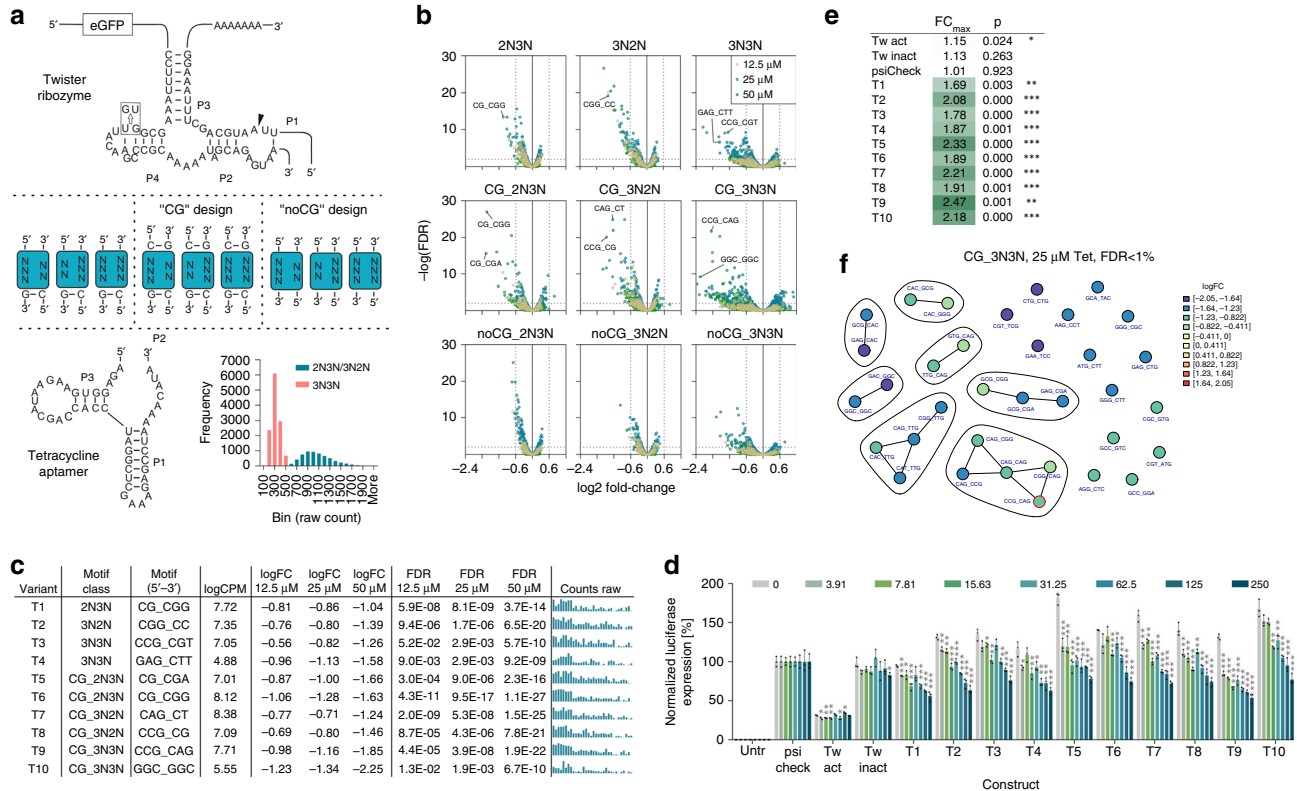

**Fig. 5 Identification of Tet-responsive Twister ribozymes by amplicon-seq. a** Secondary structure of the Tet-Twister library design. Blue boxes indicate the randomized motifs of the nine sub-libraries, connecting the Twister ribozyme and the Tet aptamer. The arrowhead indicates the cleavage site. GU/UG mutation at the indicated position renders the ribozyme inactive. Inset: Library construct frequency analysis by DNA sequencing. **b** Volcano plots displaying the significance (FDR) of changes in riboswitch abundance as a function of the change in expression (log$_2$ fold change) for the nine Tet-Twister sub-libraries. Selected hits are labeled with their sequence motif. **c** Selected hits from the Tet-Twister library screen, randomized sequence, mean counts per million (CPM), fold change (FC) and false-discovery rate (FDR) at 12.5, 25, and 50 μM Tet, respectively. The bar plots show the raw counts under untreated, 12.5, 25, and 50 μM Tet-stimulated conditions (four groups from left to right, $n = 8$ replicates each). **d** Functional hit validation in HeLa cells at increasing tetracycline doses [μM] ($n = 3$ replicates, mean ± s.d.); psi-Check: ribozyme-free control construct; Tw-act/inact: constitutively active/inactive ribozyme control. **e** Maximal fold changes and $p$-values measured upon stimulation vs. 0 μM Tet for all tested constructs. **f** Network analysis of motif similarity for the CG_3N3N library at 25 μM Tet vs. untreated and an FDR < 1%. Each line segment represents a single nucleotide change. The "T9" construct is circled in red. *$p < 0.05$, **$p < 0.01$, ***$p < 0.001$ (unpaired $T$-Test, two-tailed). Source data are provided as a Source Data file.

our screens demonstrates a high sensitivity of our method. In summary, amplicon-seq of Tet-Twister aptazymes identified the first Twister-based OFF-switches that are functional in human cells. Our method further allowed to systematically assess the effect of linker stability and symmetry on overall riboswitch function, thereby demonstrating its applicability to study RNA design features at large scale.

**Amplicon-seq enables identifying U1-snRNP-dependent switches.** In order to demonstrate the universal applicability of the introduced amplicon-seq approach, we sought to expand our method from the identification of functional riboswitches towards other RNA-based platforms, beyond aptazymes. An increasing set of synthetic riboswitch systems differing in its expression platforms has been developed in recent years[18], ranging from splicing modulation[8] to in trans systems that rely on RNA interference[19]. Prerequisites for amplicon-seq-based screening of sequence libraries based on such alternative expression platforms is a ligand-induced change in mRNA levels and an in cis regulation that allows for self-barcoding. Considering these requirements, we sought to develop a platform that is based on conditional poly-adenylation. It is known that binding of the U1 small nuclear ribonucleoprotein (U1-snRNP) upstream of a polyadenylation site

inhibits its usage and reduces mRNA levels[20,21]. Consequently, the insertion of a 9 nt long sequence (CAGGUAAGU) that is complementary to the 5′ end of the U1 snRNA, into the 3′-UTR of a CMV-*eGFP* construct, led to the effective reduction of GFP reporter expression by 7.5-fold to basal expression levels of 13.3% (U1-bs construct, see below). Fortes et al. further showed that occlusion of such a U1-snRNP binding site in a hairpin structure reverses the polyadenylation-inhibitory effect[20]. Taking these findings into account, we included the U1-snRNP binding site into a prolonged stem P1 of the guanine aptamer with the aim to place the accessibility of the U1-snRNP binding site under control of ligand binding to the aptamer (Fig. 6a and Supplementary Fig. 1e for sequence), thereby conditionally controlling polyadenylation and downstream gene expression (Fig. 6b). In this design, we randomized 6 nucleotides (Fig. 6a) and, following QC (Fig. 6c), screened the resulting library in HEK-293 cells in presence or absence of 30, 100, and 300 μM guanine.

Differential expression analysis clearly demonstrated an enrichment of ON-switches with increasing guanine-doses, as expected based on the mode of action (Fig. 6d). From the hit list ranked by log$_2$FC at 300 μM guanine (selection criteria: FDR <= 0.01, guanine dose-dependency), we selected the top ten constructs for functional validation in HEK-293 cells (Fig. 6e). The results show that all ten constructs demonstrated guanine

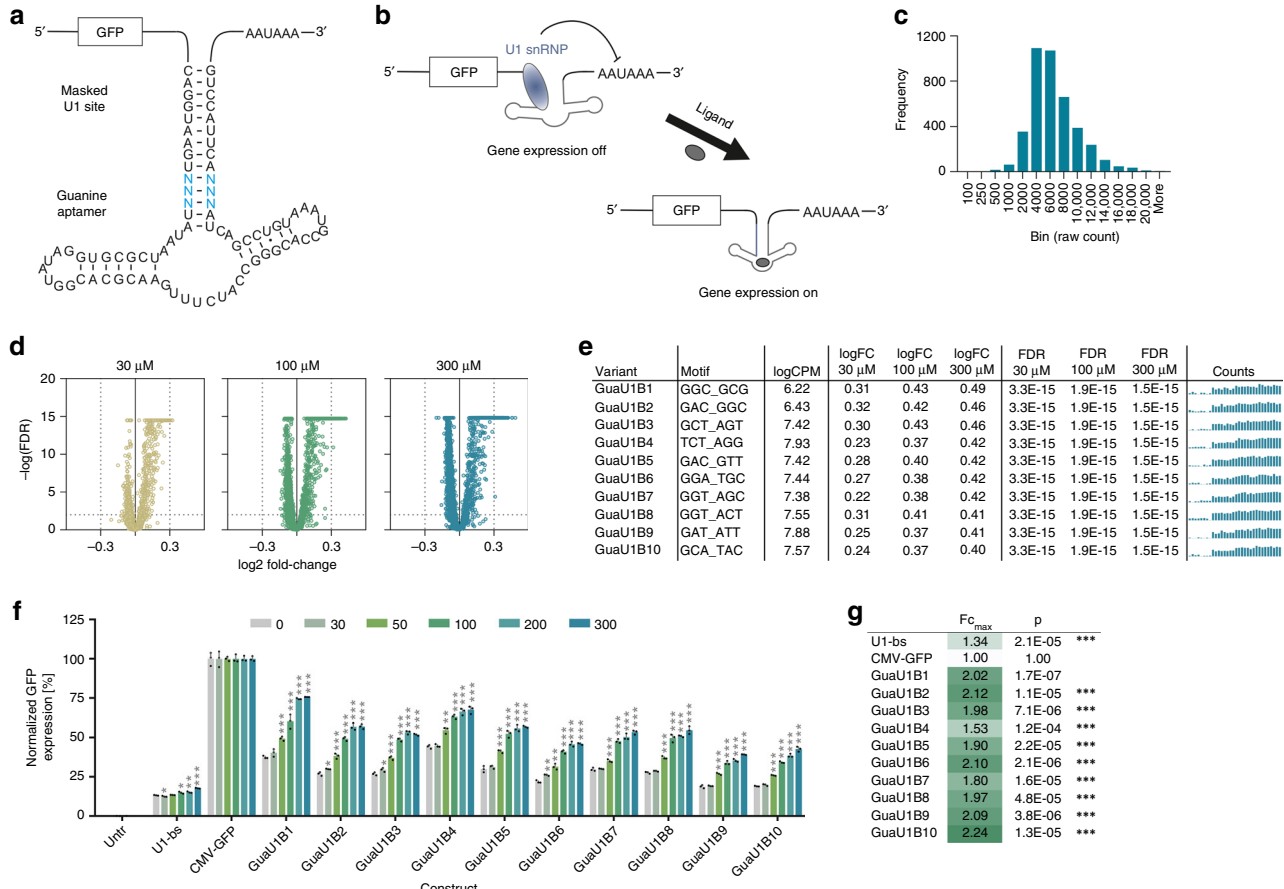

**Fig. 6 Identification of Gua-responsive riboswitches based on a U1-snRNP-dependent mode of action. a** Secondary structure of the Gua-U1-snRNP riboswitch library design. The U1-snRNP binding site (CAGGUAAGU) is included into a prolonged stem P1 of the guanine aptamer. Blue nucleotides indicate the randomized motif. **b** Mode of action of U1-snRNP-dependent riboswitches. In the ligand-unbound conformation, the U1-snRNP binding site is available, allowing for U1-snRNP binding and interference with the polyadenylation process. Upon ligand binding, the aptamer stem is stabilized and the U1-snRNP binding site is masked, enabling polyadenylation and subsequent gene expression. **c** Guanine riboswitch construct frequency analysis of the original library. **d** Volcano plots showing the significance (FDR) of changes in riboswitch abundance as a function of the change in expression ($\log_2$ fold change). **e** Selected hits from the Gua-U1-snRNP riboswitch library screen, randomized sequence, mean counts per million (CPM), fold change (FC) and false-discovery rate (FDR) at 30, 100, and 300 μM guanine, respectively. The bar plots show the raw counts under untreated, 30, 100, and 300 μM μM Gua-stimulated conditions (four groups from left to right, $n = 8$ replicates each). **f** Functional hit validation in HEK-293 cells at increasing guanine doses [μM] ($n = 3$ replicates, mean ± s.d.); U1-bs: construct harboring the freely accessible U1-snRNP binding site. **g** Maximal fold changes and $p$-values measured upon stimulation vs. 0 μM guanine for all tested constructs. $*p < 0.05$, $**p < 0.01$, $***p < 0.001$ (unpaired $T$-Test, two-tailed). Source data are provided as a Source Data file.

dose-dependent induction of gene expression (Fig. 6f). The two most interesting switches in terms of expression range and fold change were GuaU1B1 (37.5–75.7%, 2.0-fold change) and GuaU1B10 (19.1–42.7%, 2.2-fold change). These results demonstrate that our amplicon-seq method is suitable for identifying synthetic riboswitches based on alternative expression platforms, thereby significantly extending its applicability for the identification of gene expression control devices beyond aptazymes. Finally, the developed switches based on a simplified mode of action require even less coding space than the previously described aptazyme-based switches (81 nucleotides, including the ligand-binding aptamer and the expression platform), making them particularly attractive candidates for applications where expression cassette size is a limiting factor.

## Discussion

In the present work, we describe a fast, sensitive and easy-to-implement approach for the identification of artificial, ligand-responsive riboswitches in human cells in a high-throughput

manner. By applying cDNA-amplicon-seq-based counting of conditionally expressed mRNAs, functional riboswitch candidates can be rapidly identified from complex libraries containing several thousand constructs, thereby enabling the exploration of large sequence spaces. Using this method, we identified tetracycline- and guanine-inducible hammerhead and HDV riboswitches, allosterically controllable Twister-riboswitches for use in human cells and finally, switches using a mode of action based on the conditional modulation of U1-snRNP-dependent polyadenylation.

Our initial approach was based on the fact that functional aptazyme riboswitch sequences within a gene expression construct lead to the conditional, ligand-dependent cleavage and subsequent cellular degradation of the mRNA in which they are encoded. Due to the mRNA-intrinsic mode of action, the abundance and identity of riboswitch candidate sequences is directly linked to their function. A similar concept was previously applied to identify enhancer elements on a whole-genome scale, by cloning enhancer candidate libraries into reporter plasmids and quantifying the self-amplified reporter mRNAs by amplicon-seq.

This method is known as "self-transcribing active regulatory region sequencing" (STARR-seq)[22]. Recently, efforts have also been undertaken in the field of ribozyme and riboswitch engineering, to establish high-throughput screening methods. Townshend and colleagues developed a method allowing screening of construct libraries in yeast[10]. However, this method relied on FACS-selection to generate several bins of low- to high-expressing constructs prior to the sequencing-based identification of functional sequences. Moreover, their method required expensive sequence barcoding and has not yet been shown to be transferrable to human cells. More recently, the Yokobayashi group published a series of papers, utilizing sequencing of in vitro-transcribed constructs to analyze cleavage of Twister-, HDV-like drz-Agam-2-1- and Pistol ribozyme variants[23–25]. They further described a strategy for the identification of functional constructs in cells[11]. However, that method requires the isolated RNA to be separated into a non-cleaved and cleaved fraction by gel electrophoresis prior to sequencing, which again, is labor-intensive, more difficult to implement and potentially error-prone. In contrast, the method described in our work enables the rapid identification of conditionally cleaving aptazymes in a relevant cellular system, like e.g. the human cell line HEK-293, in a high-throughput manner, independent of additional pre-selection steps and costly sequence barcoding. The method relies on five core steps, namely library transfection, RNA extraction, cDNA synthesis, PCR amplification and sequencing, and enables direct screening for mRNA self-cleavage in libraries harboring up to approximately 18,000 constructs. Interestingly, while preparing our manuscript, the Smolke lab published a similar method that also successfully applies cDNA-sequencing to identify riboswitches[26]. In contrast to our method, which exploits the inherent self-barcoding nature of permutated sequence, the method described by Xiang et al. requires an additional sequence barcoding step[26]. Moreover, while both approaches underscore the applicability and value of cDNA-amplicon-sequencing for the discovery of gene switches based on ribozyme platforms, the data presented in our study further extend the use of this method towards expression platforms beyond ribozymes, as demonstrated by the U1-snRNP library screen.

A general advantage of the amplicon-seq approach is an apparent high sensitivity of detection. Many identified switches show rather low switching performance and might therefore easily be missed with alternative methods based on pre-selection, such as FACS-based techniques. Moreover, although the screening of the Tet-hammerhead- and Twister-aptazymes in the present work yielded only sequences with intermediate to low switching performances, such sequences could still be valuable, in part due to apparent differences in potency depending on the experimental context. In fact, it seems that the performance of the switches is underestimated in our setup. For example, the K19 sequence that has been rationally designed by the Süß group was originally reported to switch 4.8-fold[4]. However, we re-discovered the switch in our library and in our context the sequence switches only 2.08-fold. A likely explanation is that the sequence context alters the performance of the ribozyme switches in general. This interpretation is supported by the finding that the constitutively active hammerhead (Fig. 2g) and, even more pronounced, the active Twister ribozyme (Fig. 5d) both show considerable background in our setup, thereby likely limiting the maximum dynamic range the identified switches can achieve in our constructs. Notably, recently reported data still suggest that such switches can be useful in diverse applications, e.g., we have recently shown that the Tet K19 switch and its derivatives enable robust switching of gene expression in the nematode *C. elegans*[5]. In addition, we have demonstrated that the K19 sequence is also able to conditionally induce gene expression up to 15-fold in mice, when a secreted antibody-derivative was expressed using AAV-vector-mediated delivery of the transgene (Strobel et al., 2019, manuscript submitted). Importantly, these findings suggest that switches identified in our screen, such as the Tet-hammerhead V1 construct, which showed a 21% higher dynamic range than K19, might prove significantly more powerful in a different context, e.g. when applied to control expression in mice.

Besides Tet-hammerhead, Gua-HDV and Tet-Twister-based switches, we have also identified functional Gua-responsive hammerhead and U1-snRNP switches. The finding that powerful Gua-HHR ON-switches were identified (up to 6.9-fold induction) not only demonstrates the high value of the amplicon-seq approach for the mining of large sequence space, but also fuels hopes for the development of even higher-potency switches that would ultimately be required for therapeutic applications. Along those lines, two features of the U1-snRNP approach render it particularly attractive: First, the mode of action is simpler than that of aptazymes and second, even less coding space is required —an important characteristic, in particular in the case of AAV vectors that would profit from small and efficient gene expression control devices for future uses in gene therapy.

While a straightforward approach, several aspects requiring particular consideration to assure successful screening runs were identified during the setup of our method, which are worth to be discussed in more detail. First, constructs within a library should be available at similar abundance. This can be assured by both, monitoring library complexity during cloning/bacterial amplification of library cultures and (Sanger-) sequencing-based analysis of random samples during these process stages to estimate library complexity, and by deep sequencing analysis of the final library prior to screening run initiation. In fact, in our hands, an initial Tet-hammerhead screening run had led to a false-positive enrichment of AT-rich sequences, which retrospectively turned out to be due to a systematic underrepresentation of GC-rich sequences owing to an unintentional bias at random positions during the process of the oligonucleotide library synthesis. A GC-dependency test and graphical output (Supplementary Fig. 3) has therefore been implemented into the computational pipeline provided with this manuscript. Second, to avoid PCR amplification of riboswitch sequences from residual plasmid DNA rather than cDNA, thorough depletion of plasmid DNA during RNA isolation needs to be assured (see Methods section), while RNA integrity needs to preserved. Third, PCR should be optimized for non-saturating conditions to preserve the relative levels of each mRNA/cDNA construct. However, the necessity for PCR amplification of riboswitches from cDNA might also become obsolete by using targeted reverse transcription into cDNA (instead of random priming[26]), which will be assessed in our lab in upcoming screening runs. Fourth, when analyzing amplicon-sequencing data and selecting hits, one might encounter the well-known phenomenon that constructs of relatively low library abundance tend to show comparably high expression changes, whereas constructs with high library abundance might show highly significant p-values but lower fold changes (exemplified in Supplementary Fig. 8). It is therefore recommended to assess the sequencing counts during hit selection. The combination of fold changes and *p*-values into a common "pi-score", which was suggested to show better protection against false discoveries[27], represents an alternative selection approach (compare Supplementary Fig. 9). Finally, while we mostly observed good correlation between amplicon-seq-derived fold changes and those derived from cell culture experiments (Supplementary Fig. 10), still a few constructs behaved differently and/or turned out to be false-positive screening hits. While in case of the Tet-hammerhead run, low overall construct abundance and a less stringent FDR cutoff are likely to explain the occurrence of the

three false-positive hits, the one false-positive construct in the context of the Gua-HDV library cannot be explained. Of note, while the earliest screening run (Tet-HHR) in general showed higher fluctuations and lower correlation, the data of all following runs were significantly more robust and revealed only little (10%, Gua-HDV) or no false-positives (Gua-HHR, Tet-Twister, Gua-U1-snRNP). These findings suggest that subtle method optimization steps (e.g., using three compound doses to facilitate dose-dependency assessment), library QC and proper FDR cutoff selection help to minimize false-positive discoveries.

The method described herein offers a straightforward approach to identify functional riboswitch constructs in the human cellular context, thereby reducing the risk of limited cross-species translatability, as often observed with screening approaches in bacteria or yeast in the past[28,29]. It is conceivable that this method should also be easily transferrable to other experimental systems, including human primary cells and even in vivo models, to further increase the translational relevance or identify switches that act in a more cell-specific fashion. In particular, screening of riboswitches in mice by, e.g., AAV-mediated library expression (as already established in the context of AAV engineering[30,31]) and subsequent amplicon-seq analysis of tissue-derived RNA, has the potential to explore organ-specific riboswitch behavior, establish pharmacokinetic-pharmacodynamic relationships and therefore ultimately contribute to the identification of switches for therapeutic use.

In summary, our comparatively easy-to-implement method enables the fast and efficient identification of synthetic riboswitches from complex construct libraries, thereby overcoming one of the major hurdles on the way to therapeutically applicable switches for gene expression control.

## Methods

**Cell culture, expression plasmids, and riboswitch libraries.** HEK-293h (Thermo Fisher Scientific, cat. no. 11631017) and HeLa cells (ATCC, cat. no. ATCC-CCL-2) were cultured in Dulbecco's modified Eagle's medium (DMEM) + GlutaMAX-I + 10% fetal calf serum (Gibco/Thermo Fisher Scientific), optionally supplemented with 1% Penicillin/Streptomycin in a humidified incubator at 37 °C and 5% $CO_2$. Cells were routinely grown in standard T175 flasks (Sarstedt). The expression plasmid used for riboswitch screening and functional construct validation contained AAV2-inverted terminal repeat (ITR) sequences flanking a CMV promoter-*eGFP*-SV40-poly(A) cassette with HindIII and BglII restriction sites between the *eGFP* gene and the poly(A)-signal for riboswitch insertion. Riboswitch sequences were flanked by the following primer binding sites in 5′–3′ direction: forward: cattgcagcgtattcccagtcc, reverse: gcctggtgaaattgttatccgct. Exemplary sequences are provided in Supplementary Fig. 1. Tet-HHR and Tet-Twister riboswitch mutant libraries were synthesized at GeneArt (Thermo Fisher Scientific) based on the assembly of synthetic degenerated oligonucleotides, followed by PCR amplification, cloning into the desired CMV-*eGFP* expression plasmid backbone and amplification in *E. coli* DH10b. Library complexity and nucleotide distribution at specific positions within the randomized sequence was estimated by GeneArt, based on sequence analysis of 48 bacterial clones picked at the stage of library transformation in *E. coli*. Gua-HDV, Gua-HHR and Gua-U1-snRNP libraries were first cloned using degenerated oligonucleotides on a pET11a plasmid. After inserting the respective ribozyme and/or primer binding sites, the guanine aptamer was added by whole-plasmid PCR using a Phusion Hot Start 2 Polymerase (NEB) with primers harboring the aptamer and the respective randomized communication module in 5′ overhangs (Integrated DNA Technologies, "hand-mixing" option). The template was digested with DpnI (NEB) and the PCR product was purified by gel electrophoresis and recovered with the Zymoclean Gel DNA Recovery Kit. The DNA was circularized with Quick Ligase (NEB) and concentrated using the Clean and Concentrate Kit (Zymo Research). The constructs were then PCR amplified, cloned into the desired CMV-*eGFP* expression plasmid backbone using HindIII and BglII and transformed into *E. coli* XL10 gold (Stratagene). The bacteria were grown overnight at 37 °C on LB agar plates containing 100 μg/mL Carbenicillin. To maintain library complexity, a number of colonies exceeding 17-fold of the library size was scraped off the plates. Plasmids were isolated and sequenced at Eurofins Genomics/GATC Biotech. For Twister sequence candidate validation, sequences were cloned into the 3′-UTR of the Renilla luciferase (*hRluc*) on the psiCHECK-2 plasmid (Promega) while the firefly luciferase (hluc+) served as transfection control. After inserting a Twister ribozyme into the 3′-UTR of *hRluc*, the tetracycline aptamer was added by whole-plasmid PCR using the procedure described above. All cloning primers used in this study are listed in Supplementary Table 4.

**Screening step 1: transfection and stimulation.** Twenty-four hours before transfection, HEK-293 cells were seeded into 12-well plates at a density of 310.000 cells/well in 1 mL media. The next day (at a desired confluency of 70–80%), transfection mixtures were prepared by combining 50 μL OptiMEM media with 1.5 μL Lipofectamine-3000 (both Thermo Fisher Scientific) and another 50 μL OptiMEM with 1.2 μL P3000 and 600 ng of riboswitch library DNA. After mixing the two preparations and incubation for 10 min, 100 μL transfection mix were added to the cells. For a screening assay comprising eight replicates of each, control and three doses of ligand-treatment ($n = 32$), the amounts of all reagents and DNA were linearly upscaled (e.g., 19.2 μg DNA were used). In cases were large volumes of library DNA had to be used because of low concentrations, the volume of OptiMEM was reduced accordingly. Tetracycline (Sigma-Aldrich) was usually stored frozen as a 50 mM stock solution in light-protected tubes. Before use, it was thawed and diluted 1:10 in water (=5 mM). This solution was then used to prepare cell culture media at 12.5, 25 or 50 μM final Tet concentration. Guanine (Sigma-Aldrich) was dissolved in 0.1 M NaOH at 20 mM and diluted to the desired concentration in cell culture medium. Forty-eight hours after transfection, media was gently aspirated from the cells and Tet- or Gua-containing media was carefully added to the cells. To avoid prolonged periods without media, media was consecutively exchanged in maximally four wells at a time, going stepwise from cells receiving Tet/Gua-free to increasing doses of Tet/Gua-containing media. Stepper pipettes were used at low speed to facilitate gentle and uniform media dispensing. The cells were then incubated for three hours at 37 °C.

**Screening step 2: RNA purification.** Three hours after ligand addition, media was aspirated gently and 500 μL ice-cold phosphate-buffered saline (PBS) was pipetted onto the cells. Cells were detached by pipetting, transferred to 1.5 mL reaction tubes and placed on ice. After harvesting all wells, cells were pelleted by centrifugation at $700 \times g$ for 3 min. The supernatant was carefully but thoroughly aspirated. 370 μL RLT buffer (Qiagen) were then added to the cells, vortexed and incubated at RT for 3–5 min to facilitate lysis. After centrifugation of the cell lysates for 3 min at $10,000 \times g$, 300 μL supernatant of each sample were thoroughly mixed with 300 μL 70% ethanol and transferred to an RNeasy 96 plate (Qiagen). RNA was purified according to the manufacturer's instructions, including the optional on-column DNase step with an incubation for 25 min at RT, as described in the RNeasy 96 kit manual. RNA was finally eluted by two consecutive elution steps, using 45 μL RNAse-free water each. A 10-minute in-solution DNase step was carried out to deplete residual DNA as per protocol (see Qiagen RNeasy MinElute Cleanup kit manual), followed by RNA cleanup as per RNeasy 96 kit protocol. In general, whenever possible, multichannel or stepper pipettes were used to facilitate speedy RNA purification and decrease the risk of RNA degradation. RNA concentration was determined spectrophotometrically, using NanoDrop devices.

**Screening steps 3–4: reverse transcription, PCR amplification.** One microgram of each RNA sample was then used as a template for reverse transcription into cDNA, using the High capacity cDNA Reverse Transcription Kit (Applied Biosystems, Thermo Fisher Scientific) and a reaction volume of 40 μL. In addition, non-transcription control reactions that lacked the reverse transcriptase were prepared. Following the instructions of the kit manual, reverse transcription was conducted in a PCR cycle for 10 min at 25 °C, 120 min at 37 °C and 5 min at 85 °C. After cooling down the samples to 4 °C, 1 μL of each reaction (usually corresponding to approximately 2.3 μg cDNA, as assessed per Nanodrop) was transferred to a PCR plate for riboswitch sequence amplification, using following primers: RS-ampl-forward (5′3′): cattgcagcgtattcccagtcc, RS-ampl-reverse (5′3′): agcggataacaatttcaccaggc. For each reaction, the cDNA was mixed with 4 μL 5× Phusion Buffer HT (Thermo Fisher Scientific), 0.4 μL dNTPs (10 mM stock), 1 μL of both, forward and reverse primers (10 μM stock), 0.5 μL Phusion Enzyme (2U/μL) and 12.1 μL $H_2O$. Because a usual experiment comprised 32 samples, the PCR mix was usually set up as a 35× master mix. PCR conditions were as follows: 30 s at 98 °C, followed by 22 cycles of 5 s at 98 °C, 10 s at 68 °C and 15 s at 72 °C, followed by a final elongation step at 72 °C for 7 min and subsequent cooling at 10 °C. As a negative control, several RNA samples that were not reversely transcribed were also PCR amplified to prove they were free of residual plasmid DNA. PCR reactions were finally purified using the NucleoSpin Gel and PCR Clean-up kit (Macherey-Nagel) and eluted in 20 μL $H_2O$. Higher-throughput PCR purification was carried out using the MagMax Express-96 Magnetic Particle processor and AMPure Beads from Beckmann Coulter according to the manufacturer's clean-up protocol.

**Screening step 5: cDNA-amplicon-seq and hit selection.** PCR product DNA concentration was determined on the Fragment Analyzer with NGS Standard Sensitivity according to manufacturer's instructions. Subsequently, end repair was carried out using the Ovation Library System for Low Complexity Samples kit (NuGEN) with the manufacturer's protocol, followed by adaptor ligation from the same kit. The libraries underwent cleanup again using the MagMAX Express-96 Magnetic Particle processor and the Agencourt RNAClean XP beads (Beckmann Coulter) following the manufacturer's protocol. Subsequently, library amplification was carried out using the Ovation Library System for Low Complexity Samples kit (NuGEN) according to the manufacturer's protocol, followed by a clean-up step

with Agencourt AMPure XP Beads (Beckmann Coulter). The libraries were quantified using a Quant-iT PicoGreen dsDNA Assay kit (Invitrogen) according to the manufacturer's instructions and fragment size was determined with the NGS Standard Sensitivity Kit (Advanced Analytical) on a Fragment Analyzer. Based on the concentrations obtained from the PicoGreen measurement, a 2 nM dilution of the libraries in Illumina Resuspension Buffer + 0.1% Tween20 preparation (Sigma) was prepared and in the following, 10 µL of each sample were mixed with a unique reverse adaptor from the 2 nM dilution. Libraries were denatured by consecutive addition of 10 µL 0.2 N NaOH, µL of 200 mM Tris-HCl, pH 7.0 and 970 µL of prechilled HT1 buffer. In a new prechilled tube, 1183 µL of prechilled HT1 buffer were added. Subsequently, 117 µL of the library pool dilution and 2 µL of 20 pM PhiX control were added. Finally, the libraries were sequenced on an llumina NextSeq 500, using 10 million single-end reads of 154 bp length per sample, using the NextSeq 500/550 High Output Kit v2.

**Screening step 6: bioinformatic analyses**. A program built in-house was used to count the abundance of the different aptamer variations found in the rawdata (in FASTQ format) from each library. The program searches for a set of patterns using pre-defined regular expressions (see Supplementary Table 5 for an example) and reports the sequence abundance of each variable region (random pool) in the aptamer along with various quality control read-outs (e.g., the GC-dependence of the abundance distribution, Supplementary Fig. 3). Sequence abundance was then normalized with respect to a spike-in sequence present in the same concentration in all libraries. The edgeR[32] package in R[33] was used to identify significant changes in pattern abundance in the different treatment conditions, and for summarizing the results in volcano plots for each of the contrasts (Supplementary Fig. 4). The sequences of significantly changed variants were further subjected to a variety of sequence similarity analyses. For this, similar sequences were first identified by calculating the Hamming distance between them, and calculating the hierarchical cluster on those distances, cutting the tree at a height of 0.75 (which would group sequences with >75% identity) (Supplementary Fig. 5a). By using the package motifStack[34], a position weight matrix (PWM) for each of those clusters was calculated and used to generate motif logos for the individual clusters, given a big enough size (Supplementary Fig. 5b). Hamming distances were further used to build a network: an edge was defined between two nodes (significant variable sequences) when the distance between the sequences was only 1 nucleotide. With this, we created an adjacency matrix and identified communities of nodes based on the edge betweenness algorithm (igraph[35]), identifying trends in the direction of fold changes within and between communities (Supplementary Fig. 6). Predicted fold structure of every sequence and minimum free energy (MFE) of the predicted structure was calculated using RNAfold 2.3.5 of the ViennaRNA package[36]. A complete description of the pipeline can be found in the README.md available in GitHub.

**Validation of construct functionality in cell culture**. Selected sequences of potential hit constructs were tested for their ability to regulated reporter gene expression in eGFP or dual-luciferase assays in HEK293 and HeLa cells, respectively. For Tet-hammerhead and Gua-riboswitch construct validation, 25,000 HEK293 cells were seeded per well of a 96-well plate the day before transfection. Using the Lipofectamine-3000 kit (Thermo Fisher Scientific), cells were transfected with 35 ng of plasmid DNA and subsequently supplied with stimulation media containing increasing amounts of tetracycline. In case of Gua-responsive constructs, cells were calcium-phosphate transfected by mixing 70 ng DNA with 10 µL 300 mmol/l CaCl$_2$ and 10 µL 2× HBS buffer (50 mmol/l HEPES, 280 mmol/l NaCl, 1.5 mmol/l Na$_2$HPO$_4$) and adding 20 µL of this mix per well of a 96-well plate[3]. The media was changed to Gua-containing media approximately 4 h after addition of the transfection mix. 24 h after transfection, eGFP fluorescence was measured using a Molecular Devices SpectraMax i3x fluorescence reader. For Twister ribozyme validation, 20,000 HeLa cells per well were seeded in a 96-well plate. The next day, plasmids were transfected into the cells using Lipofectamine 3000 according to the manufacturer's instructions. After 4 h, medium was exchanged to medium with or without Tet. 24 h after transfection, the Dual-Glo Luciferase Assay (Promega) was performed. The activity of the hRluc was normalized to the activity of the hluc.

**Statistics**. Statistical calculations were performed using GraphPad Prism V7.03 and R version 3.3.2 (2016-10-31).

**Reporting summary**. Further information on research design is available in the Nature Research Reporting Summary linked to this article.

## Data availability
The data that support the findings of this study are available from the corresponding author upon reasonable request. The raw sequencing data for the riboswitch screens have been deposited in NCBI's Gene Expression Omnibus and are accessible through GEO Series accession number GSE143466 [https://www.ncbi.nlm.nih.gov/geo/query/acc.cgi?acc=GSE143466]. The source data underlying Figs. 2b–g, 3b–e, 4b–e, 5a–d and 6c–f are provided as a Source Data file.

## Code availability
The full computational processing pipeline is available via GitHub at [https://www.github.com/bi-compbio/riboswitch-pipeline/].

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

## Acknowledgements

The authors thank Tobias Hildebrandt, Coralie Viollet and Dagmar Knebel for supporting amplicon-sequencing and Astrid Joachimi (University of Konstanz) for excellent technical assistance.

## Author contributions
B.S., M.S., J.S.H. and S.K. conceived the study and designed all experiments. B.S., M.S., D.B. and W.R. performed and analyzed experiments. H.K. and S.P. developed the computational analysis pipeline and conducted additional customized analyses. B.S., H.K. and S.K. wrote the manuscript draft. All authors edited and contributed to the final version of the manuscript.

## Competing interests

All authors listed, except from M.S. and J.S.H., are employees of Boehringer Ingelheim Pharma GmbH & Co. KG.
