## [Peer Review File · Nature Communications]

Reviewers' Comments:

Reviewer #1:

Remarks to the Author:

The authors describe a method for optimizing designs of aptazyme communication modules, based on differential Illumina read count for full-length (uncleaved) transcripts in the presence of varying analyte levels. In brief, analyte binding to an aptamer module drives a conformation change in the (Hammerhead or Twister) self-cleaving ribozyme domain that either activates (Hammerhead) or suppresses (Twister) self-cleavage. Cleaved mRNAs are degraded. Because PCR amplification crosses the cleavage site, only uncleaved mRNAs contribute to Illumina reads, so that this is a loss-of-signal assay. Importantly for this design, the variant sequence responsible for the behavior of a given transcript are embedded within the sequence information obtained for that transcript. The design can handle approximately 200k variants. This is a good number for evaluating relatively small libraries (5-7 randomized positions).

The work is well designed and carefully implemented. It is a nice refinement on existing methodologies and an overall positive contribution that should be published. However, neither the magnitude nor the innovativeness of the work are appropriate for Nat Comm.

The primary innovations here are 1) selection design, including the reliance on read count and the self-reporting feature noted above; and 2) application of this design to Twister ribozyme within the context of human (HEK293) cells. This group has already done similar work in *E. coli* and *S. cerevisiae*, and other groups have used (admittedly more cumbersome) *in vivo* selection methods for ligand-gated Twister allostery in mammalian cells. New insights are present (but sparse and incremental) regarding ribozymes, allostery, RNA folding landscapes, selection methods, informatics and RNA biology.

Discussion claims that "the identification of such low-performing switches might be advantageous," and then speculates about several contexts where the authors anticipate there to be advantages. This claim and the speculations are a stretch. The repeated assertions that there could be utility in the weak signal are unsubstantiated. While speculation is certainly allowable, the assertions do not make the case that the advantages exist. For example, it is unclear why a low performer would be better than a high performer for screening novel aptamers. It would be more compelling to demonstrate such advantages. More likely that this is an over-reach.

Discussion paragraph beginning, "While a straightforward approach, several aspects requiring particular consideration" offers several points of potentially useful guidance for implementing the methodology, although much of it is in the form of appendix (not carefully addressed by the data presented here, such as the quantitative impact of initial frequency distributions) or truisms (confounding effect of carry-over plasmid DNA). This paragraph is only mildly useful.

Last long paragraph of Discussion is not needed. Could be simplified to one sentence noting that the essential design could be extended to primary cells and to whole animal systems.

MINOR POINTS

Abstract, line 3. Suggestion: "...due to the limited understanding of context-dependent structure-function relationships"

p3 line 6. Use different verb. "Impacting" is ambiguous here (e.g., could mean "stabilize" or "destabilize"). Four lines later, change "the most" to "an" (no need to pick a fight; they are clearly attractive).

p4 line 4. Fix "reversely transcribed" (omit "ly"). Last two lines—Suggestion for simplification: "Finally, the computational analysis pipeline enables comprehensive analyses..."

p6. Unclear meaning in two places: "cells were lysed and RNA was purified using column purification" (oligo-dT beads? size exclusion? activated silica?) and "by first normalizing all constructs' counts to a control plasmid spiked into the library" (was control plasmid co-transfected? Unclear whether authors are looking at mRNA from that plasmid or at plasmid DNA. – very important distinction!)

Fig 2D legend ambiguity. What is on the x-axis and what is on the y-axis? e.g., is "associated fold changes reported by Beilstein et al." on x- or y-axis? Is "fold changes measured in our screen" on x- or y-axis?

Fig 2E legend ambiguity. Specify whether the tiny bar graph data are in eight groups of three or three groups of eight.

p10 bottom and p11 top, several sentences included phrases such as "The data shows" and "The data suggests." Don't do this. Point to the specific observations in the data that lead to these conclusions, THEN state the inferences you are deriving from them. (Also, DATUM is singular, DATA is plural – "The data ARE..."). Later in same paragraph, capitalize Hamming (person's name).

Fig 5 legend, point out that each line segment represents a single nucleotide change.

Discussion, 2nd paragraph. "... construct LED to the conditional, ligand-dependent cleavage and subsequent cellular degradation of the mRNA IN WHICH they are encoded."

p16 line 4, change PROOF to PROVE. Middle of same page: "has therefore BEEN implemented". Next page at end of this same long paragraph: "show better protection AGAINST false discoveries"

Reviewer #3:

Remarks to the Author:

In this manuscript, Strobel et al. described the method for screening synthetic riboswitches in human cells. They obtained tetracycline-responsive ON/OFF switches that have high expression fold-change rather than that of previously reported. It is an interesting approach to identify synthetic riboswitches. However, there are several concerning points about the current manuscript.

Major comments

1. The authors only used tetracycline aptamer. I wonder whether this strategy can be applied for other ligands-responsive aptamers or not. There are so many tetracycline responsive aptazymes, thus I could not find the outstanding leap compared with previous studies except screened in human cells.

2. The authors checked the expression-fold change of riboswitches that were highly ranked by their NGS-based method. However, to demonstrate that the NGS-based method can efficiently extract truly functional riboswitches, the authors should compare the frequency of highly active riboswitches in the high-ranked group and randomly selected (or low-ranked) groups by an expression-based assay. Without such comparison, it is difficult to say that the NGS-based method can efficiently extract functional riboswitches especially because the correlation of the rank in NGS-based method and actual riboswitch activities seems not to be so high in figure 2e-g.

3. The authors showed a correlation between the fold change of their NGS and reported fold change based on the expression ratio. However, the authors should also show a correlation between NGS data and expression validations of their extracted riboswitches.
4. The authors said 'The three non-functional constructs displayed strongly reduced basal eGFP expression' but why such false positive occurred? Some of them showed high FDR, but the reason of the technical problem should be explained.
5. The authors mentioned Twister ribozyme can serve as an expression platform for novel aptazymes with potentially higher potency. But in their results, the fold change of expression level is not as good as previous tetracycline responsive OFF switches that used other type ribozymes.
6. In fig5, the authors used HeLa cells and Luciferase instead of HEK293 and GFP. Why?
7. In line 5 of page 8, the authors said that they removed the constructs with an AT-content >70% (that was predicted to disturb overall riboswitch structure). However, in V12 (GTATT_AA), which is included in the remaining 14 constructs, 6 of 7 nucleotides is A or T (AT-content = 86%). Why the authors did not remove it? In addition, V12 showed a significant change in GFP expression in figure 2f. If the construct with an AT-content >70% like V12 can be active, the authors should check the activity of other high-rank constructs with an AT-content >70% (e.g., TAAAC_AG).
8. If FDR is less than 1, $\log(\text{FDR})$ should be less than 0 (e.g., if $\text{FDR} = E-8$, $\log(\text{FDR}) = -8$), and $-\log(\text{FDR})$ should be larger than 0. However, while FDRs shown in figure 4b are less than 1, $-\log(\text{FDR})$ shown in figure 4a is less than 0. Is the vertical axis correct?
9. In luciferase assay, the authors used hluc to compensate for the difference of transfection efficiency and normalize hRluc expression. However, there is no description of how they compensated the difference of transfection efficiency in GFP assay. Did they co-transfect a gene of another fluorescent protein to normalize GFP expression?

Minor comments

1. There are two kind of fold change (NGS and expression level), this is confusing.
2. The abbreviation are not explained enough. (e.g. ITR)
3. In figure 4a, the dots of 12.5 μM is hard to see. In addition, the difference in colors of 25 and 50 μM dots are not so apparent. I recommend the authors to use totally different colors in each group or show the dots of each concentration in different plots.
4. The authors should submit figures with higher resolution. Especially for figure 2c and 4a, because the "mosquito noises" make it difficult to distinguish each dot.

Response to the reviewers

We would like to thank the reviewers for their helpful comments and the editors for their invitation to prepare a revised version of the manuscript. We have used this opportunity to significantly extend our work by providing additional application examples for the identification of RNA switches based on barcode-free amplicon-sequencing. Specifically, we designed and screened two additional aptazyme riboswitch libraries, including one additional ribozyme and one additional aptamer. Moreover, in order to further increase the novelty and to demonstrate the general applicability of our work, we introduce a new mode of action for aptamer-based control of gene expression by regulating U1-snRNP-modulated polyadenylation of a transcript. By applying our screening approach, we were able to identify several switches of gene expression that function via this simplified mechanism, demonstrating the broad applicability of our screening approach, even beyond ribozyme-based mechanisms. During the revision phase, a similar method was published in this journal by the Smolke group (Xiang et al. Nat Commun 2019). We briefly discuss this latest work in the revised manuscript. From our perspective, the two main differences between the two studies lie in the use of sequence barcoding and the restriction to aptazyme switches in the Xiang paper, whereas our method exploits the self-barcoding nature of library constructs and further demonstrates applicability beyond aptazyme-based library designs. Based on these aspects, we are convinced that our manuscript will be a valuable addition to the field.

Reviewers' comments:

Reviewer #1 (Remarks to the Author):

The authors describe a method for optimizing designs of aptazyme communication modules, based on differential Illumina read count for full-length (uncleaved) transcripts in the presence of varying analyte levels. In brief, analyte binding to an aptamer module drives a conformation change in the (Hammerhead or Twister) self-cleaving ribozyme domain that either activates (Hammerhead) or suppresses (Twister) self-cleavage. Cleaved mRNAs are degraded. Because PCR amplification crosses the cleavage site, only uncleaved mRNAs contribute to Illumina reads, so that this is a loss-of-signal assay. Importantly for this design, the variant sequence responsible for the behavior of a given transcript are embedded within the sequence information obtained for that transcript. The design can handle approximately 200k variants. This is a good number for evaluating relatively small libraries (5-7 randomized positions).

The work is well designed and carefully implemented. It is a nice refinement on existing methodologies and an overall positive contribution that should be published. However, neither the magnitude nor the innovativeness of the work are appropriate for Nat Comm.

We thank the reviewer for acknowledging the quality and value of our study. As detailed further below, we now incorporated several new screening data sets that strongly extend the overall data set to new ribozymes and a further ligand (new Figures 3 and 4) and hence confirm the broad applicability of our method. Several of these new switches show very promising activity (e.g. a 6.9-fold expression-inducing Guanine-hammerhead ON-switch) Moreover, we now introduced a novel mode of riboswitch action based on the conditional occupation of a U1-snRNP binding site, thereby controlling polyadenylation. By demonstrating applicability of our method for the identification of switches based on this innovative mechanism (new Fig. 6), the method's broad utility is additionally underscored. We hope that the extensive and innovative additions to our manuscript will convince Reviewer #1.

The primary innovations here are 1) selection design, including the reliance on read count and the self-reporting feature noted above; and 2) application of this design to Twister ribozyme within the context of human (HEK293) cells. This group has already done similar work in *E. coli* and *S. cerevisiae*, and other groups have used (admittedly more cumbersome) in vivo selection methods for ligand-gated Twister allostery in mammalian cells. New insights are present (but sparse and incremental) regarding ribozymes, allostery, RNA folding landscapes, selection methods, informatics and RNA biology.

Discussion claims that “the identification of such low-performing switches might be advantageous,” and then speculates about several contexts where the authors anticipate there to be advantages. This claim and the speculations are a stretch. The repeated assertions that there could be utility in the weak signal are unsubstantiated. While speculation is certainly allowable, the assertions do not make the case that the advantages exist. For example, it is unclear why a low performer would be better than a high performer for screening novel aptamers. It would be more compelling to demonstrate such advantages. More likely that this is an over-reach.

We agree with the reviewer and apologize for our initial argumentation that was somewhat misleading. Our statements were not meant to justify the value of low-performing switches, but rather to point out that a) even low-performing switches can be identified, thereby demonstrating the sensitivity of our method and b) that switch performance might be strongly dependent on the experimental system used. As an example, we mentioned the K19 switch that showed a dynamic range of ~ 5-fold in HeLa cells in the paper by Beilstein et al., 2.1-fold in our hands in HEK-cells, but ~ 15-fold in a mouse study carried out in our lab. We therefore concluded that the low-performing switches might still be of value in different experimental settings, which we believe is a valid statement. Our thought regarding the statement on aptamer identification was that the application of novel aptamers might initially only result in weakly performing switches but that such sequences are nevertheless entry points for further optimization. Here, a sensitive assay would also be of value. However, because this question is clearly beyond the scope of our manuscript and, in order to address the concerns raised by the reviewer, we deleted the respective paragraph in the revised manuscript.

Discussion paragraph beginning, “While a straightforward approach, several aspects requiring particular consideration” offers several points of potentially useful guidance for implementing the methodology, although much of it is in the form of appendix (not carefully addressed by the data presented here, such as the quantitative impact of initial frequency distributions) or truisms (confounding effect of carry-over plasmid DNA). This paragraph is only mildly useful.

We thank the reviewer for this hint. While we understand that some of the raised aspects might in fact seem obvious, it was our clear intention to provide all practical aspects that we experienced to be of importance when setting up the method, in order to guide the setup in other labs. However, to comply with the reviewer’s suggestion, we now shortened the paragraph.

Last long paragraph of Discussion is not needed. Could be simplified to one sentence noting that the essential design could be extended to primary cells and to whole animal systems.

We now also shortened that paragraph.

MINOR POINTS

Abstract, line 3. Suggestion: "...due to the limited understanding of context-dependent structure-function relationships"

p3 line 6. Use different verb. "Impacting" is ambiguous here (e.g., could mean "stabilize" or "destabilize"). Four lines later, change "the most" to "an" (no need to pick a fight; they are clearly attractive).

p4 line 4. Fix "reversely transcribed" (omit "ly"). Last two lines—Suggestion for simplification: "Finally, the computational analysis pipeline enables comprehensive analyses..."

We thank the reviewer for these suggestions, which we incorporated accordingly.

p6. Unclear meaning in two places: "cells were lysed and RNA was purified using column purification" (oligo-dT beads? size exclusion? activated silica?) and "by first normalizing all constructs' counts to a control plasmid spiked into the library" (was control plasmid co-transfected? Unclear whether authors are looking at mRNA from that plasmid or at plasmid DNA. – very important distinction!)

We now changed the wording to be more specific. The sections now read: "... cells were lysed and RNA was purified using silica column purification" and "by first normalizing all constructs' cDNA amplicon counts to a control plasmid co-transfected with the library".

Fig 2D legend ambiguity. What is on the x-axis and what is on the y-axis? e.g., is "associated fold changes reported by Beilstein et al." on x- or y-axis? Is "fold changes measured in our screen" on x- or y-axis?

Fig 2E legend ambiguity. Specify whether the tiny bar graph data are in eight groups of three or three groups of eight.

We apologize for not being clear. We changed the figure legend to now clearly specify these points.

p10 bottom and p11 top, several sentences included phrases such as "The data shows" and "The data suggests." Don't do this. Point to the specific observations in the data that lead to these conclusions, THEN state the inferences you are deriving from them. (Also, DATUM is singular, DATA is plural – "The data ARE..."). Later in same paragraph, capitalize Hamming (person's name).

We thank the reviewer for these suggestion and incorporated the changes accordingly. Specifically, the section now reads: "Interestingly, the observation that higher fold changes were observed for all 3N3N libraries compared to 2N3N and 3N2N designs further suggests that the symmetric communication module design leads to switches of higher potency than the asymmetric design. Moreover, the fraction of constructs displaying higher fold changes and/or higher significance was biggest in the "CG"-sub-libraries, suggesting that linker stabilization by the Twister-proximal CG pair has an overall beneficial effect."

Fig 5 legend, point out that each line segment represents a single nucleotide change.

Discussion, 2nd paragraph. "... construct LED to the conditional, ligand-dependent cleavage and subsequent cellular degradation of the mRNA IN WHICH they are encoded."

p16 line 4, change PROOF to PROVE. Middle of same page: "has therefore BEEN implemented". Next page at end of this same long paragraph: "show better protection AGAINST false discoveries"

We again incorporated the suggested changes accordingly.

Reviewer #2 (Remarks to the Author):

In this manuscript, Strobel et al. described the method for screening synthetic riboswitches in human cells. They obtained tetracycline-responsive ON/OFF switches that have high expression fold-change rather than that of previously reported. It is an interesting approach to identify synthetic riboswitches. However, there are several concerning points about the current manuscript.

We thank the reviewer for the interest in our manuscript and are very confident that the additionally included data and changes as per her/his suggestion substantially improved our manuscript.

Major comments

1. The authors only used tetracycline aptamer. I wonder whether this strategy can be applied for other ligands-responsive aptamers or not. There are so many tetracycline responsive aptazymes, thus I could not find the outstanding leap compared with previous studies except screened in human cells.

We thank the reviewer for this valid question. On top of the previously included Tet-hammerhead and Tet-Twister data, we have now included data for guanine-HDV (new Fig. 3), guanine-hammerhead (new Fig. 4), and guanine-U1-snRNP riboswitch designs (new Fig. 6), thereby strongly broadening our dataset and clearly demonstrating the broad utility of our method. Several of the newly identified riboswitch candidates demonstrate high potency and therefore are promising candidates for further engineering efforts. Importantly, with introducing a novel concept for RNA-based switches of gene expression based on the ligand-dependent control of the accessibility of a U1-snRNP binding site in the 3'-UTR, we hope that we can convince reviewer 2 that the present study gained significantly in novelty and general interest.

2. The authors checked the expression-fold change of riboswitches that were highly ranked by their NGS-based method. However, to demonstrate that the NGS-based method can efficiently extract truly functional riboswitches, the authors should compare the frequency of highly active riboswitches in the high-ranked group and randomly selected (or low-ranked) groups by an expression-based assay.

Without such comparison, it is difficult to say that the NGS-based method can efficiently extract functional riboswitches especially because the correlation of the rank in NGS-based method and actual riboswitch activities seems not to be so high in figure 2e-g.

We thank the reviewer for this suggestion. Following this idea, we have tested the performance of low-ranking switch constructs, which had not fulfilled our hit selection criteria (specifically, those on ranks #500, #1000, #2000 and #3000). As expected, the data show that these switches, as opposed to the high-ranking ones, are non-functional. The respective data has now been included as the new Supplemental Fig. 7 and a respective statement has now been included in the results, which reads:

The observation that all ten sequences selected for cellular characterization showed Tet-dose-dependent control of gene expression demonstrates that the screening approach together with the described selection criteria results in a low false-positive hit discovery rate. This notion is further supported by the finding that four randomly selected constructs that did not fulfill our selection criteria (i.e., they were found on ranks #500, #1000, #2000 and #3000 of a negative selection list based on a log₂-fold change >-0.6 and an FDR>0.01 at 300 μM) did not show switching behavior, as expected (Supplementary Figure 7).

3. The authors showed a correlation between the fold change of their NGS and reported fold change based on the expression ratio. However, the authors should also show a correlation between NGS data and expression validations of their extracted riboswitches.

We thank the reviewer for this suggestion. We have now incorporated respective graphs for all screening runs in Supplemental Figure 10. We have further addressed the aspect of sequencing- and validation experiment-derived fold changes together with the question of false-positives in a dedicated paragraph in the discussion, which reads:

Finally, while we mostly observed good correlation between amplicon-seq-derived fold changes and those derived from cell culture experiments (Supplementary Fig. 10), still a few constructs behaved differently and/or turned out to be false-positive screening hits. While in case of the Tet-hammerhead run, low overall construct abundance and a less stringent FDR cutoff are likely to explain the occurrence of the three false-positive hits, the one false-positive construct in the context of the Gua-HDV library cannot be explained. Of note, while the earliest screening run (Tet-HHR) in general showed higher fluctuations and lower correlation, the data of all following runs were significantly more robust and revealed only little (10%, Gua-HDV) or no false-positives (Gua-HHR, Tet-Twister, Gua-U1-snRNP). These findings suggest that subtle method optimization steps (e.g., using three compound doses to facilitate dose-dependency assessment), library QC and proper FDR cutoff selection help to minimize false-positive discoveries.

4. The authors said 'The three non-functional constructs displayed strongly reduced basal eGFP expression' but why such false positive occurred? Some of them showed high FDR, but the reason of the technical problem should be explained.

We can only speculate that the identification of false-positives might occur because of low overall construct abundance within a library (where even small changes that rather originate from noise

result in significant fold changes), overall experimental fluctuations and insufficient FDR cutoffs chosen during hit selection. We have discussed these aspects in the revised version (see reviewer comment 3 above).

5. The authors mentioned Twister ribozyme can serve as an expression platform for novel aptazymes with potentially higher potency. But in their results, the fold change of expression level is not as good as previous tetracycline responsive OFF switches that used other type ribozymes.

We agree with the reviewer's interpretation. We stated that the use of the Twister ribozyme could potentially yield even better switches since we found this to be true, based on previous findings in the context of E. coli (Felletti et al. Nat Commun, 2016). However, using the current design, this finding seemingly does not translate to human cells. Accordingly, we have now removed this statement from the manuscript.

6. In fig5, the authors used HeLa cells and Luciferase instead of HEK293 and GFP. Why?

The use of different cell lines and reporter systems originates from the utilization of the already established systems that differed in the Hartig and BI labs. However, the use of different reporters and human cell lines demonstrates the general functionality of the approach independent of these parameters.

7. In line 5 of page 8, the authors said that they removed the constructs with an AT-content >70% (that was predicted to disturb overall riboswitch structure).

However, in V12 (GTATT_AA), which is included in the remaining 14 constructs, 6 of 7 nucleotides is A or T (AT-content = 86%). Why the authors did not remove it?

In addition, V12 showed a significant change in GFP expression in figure 2f. If the construct with an AT-content >70% like V12 can be active, the authors should check the activity of other high-rank constructs with an AT-content >70% (e.g., TAAAC_AG).

We thank the reviewer for this hint. In fact, V12 was kept, as it harbored the initial GTA motif that seemed to be enriched among the top hits and was not predicted to disrupt ribozyme folding. As per the reviewer's suggestion, we now tested the three highest-ranking AT-rich constructs, which (in accordance with our prediction) were not functional. We provide these data for review purposes only (see below) and maintain our initial argumentation in the manuscript. However, as we had omitted to explain why V12 had been kept, we adapted the respective sentence, which now reads:

After removal of three duplicate hits and three of four constructs with an AT-content >70% (that was predicted to disturb overall riboswitch structure (construct V12 was kept due to the GTA-motif that seemed to be enriched among the selected hits)), 14 constructs remained and were applied to functional validation in HEK-293 cells.

8. If FDR is less than 1, $\log(\text{FDR})$ should be less than 0 (e.g., if $\text{FDR} = E^{-8}$, $\log(\text{FDR}) = -8$), and $-\log(\text{FDR})$ should be larger than 0. However, while FDRs shown in figure 4b are less than 1, $-\log(\text{FDR})$ shown in figure 4a is less than 0. Is the vertical axis correct?

We thank the reviewer for carefully proofreading and pointing this mistake out to us. The axis labeling has now been corrected.

9. In luciferase assay, the authors used hluc to compensate for the difference of transfection efficiency and normalize hRluc expression. However, there is no description of how they compensated the difference of transfection efficiency in GFP assay. Did they co-transfect a gene of another fluorescent protein to normalize GFP expression?

In GFP assays, we do not usually see big differences in transfection efficiency (when using Lipofectamine transfection in conjunction with multichannel cell seeding etc.), which is why we do not routinely include co-transfected reporters. However, the combined plate reader and microscope device used to measure GFP direct fluorescence also assesses the ratio of GFP-positive cells per well. We routinely check this data to identify strong well-to-well fluctuations, which however, were not observed in any of the experiments shown in this paper.

Minor comments

1. There are two kind of fold change (NGS and expression level), this is confusing.

We now adapted the wording wherever relevant to be clearer, specifically in the legend of Fig. 2e.

2. The abbreviation are not explained enough. (e.g. ITR)

We now included an explanation (inverted terminal repeat).

3. In figure 4a, the dots of 12.5 μ M is hard to see. In addition, the difference in colors of 25 and 50 μ M dots are not so apparent. I recommend the authors to use totally different colors in each group or show the dots of each concentration in different plots.

We now revised these figures and changed the colors and appearance, to make them clearer.

4. The authors should submit figures with higher resolution. Especially for figure 2c and 4a, because the "mosquito noises" make it difficult to distinguish each dot.

We did, however, the figures are scaled down during PDF merging during submission. High-resolution figures were now again provided along with the revised version of the manuscript.

Reviewers' Comments:

Reviewer #1:

Remarks to the Author:

Designer aptazymes could be useful for a wide range of applications, but engineering novel aptazymes has proven to be a challenge. The authors describe a method for optimizing designs of aptazyme communication modules in which analyte binding to an aptamer module drives a conformation change in the self-cleaving ribozyme domain that either activates or suppresses self-cleavage, leading to mRNAs degradation. The design can handle approximately 200k variants. This is a good number for evaluating relatively small libraries (5-7 randomized positions).

The work is well designed and carefully implemented and provides some new insights. It builds upon prior work in *E. coli* and *S. cerevisiae* in this groups and ligand-gated Twister allostery in mammalian cells in other groups. It is a nice refinement on existing methodologies and an overall positive contribution that should be published.

Reviewer #3:

Remarks to the Author:

The authors answered the most of my previous concerning points. However, there are still a few points that I should indicate.

Major comments

1. Regarding my previous major comment 2, the authors now show that low-ranking Tet-Twister ribozymes have lower performance than high-ranking Tet-Twister ribozymes in new Supplemental Fig. 7. However, in other switches (hammerhead ribozymes, HDV ribozymes, and Gua-U1-snRNP riboswitches), the superiority of high-ranking constructs over low-ranking constructs was not confirmed. The data of low-ranking constructs in other switches will make the study more persuasive.

Minor comments

1. Regarding my previous major comment 8, the vertical axis of the previous figure 4a (new figure 5b) was properly corrected. However, there is still the same mistake in the vertical axis of the new figure 6d.

2. While the variant names of Gua-responsive riboswitches in figure 6e-g are described as "GuaU1B1" and "GuaU1B10", in page 18 line 379-380, these variants are described as "GuaB1U1" and "GuaB10U1", respectively.

3. Related to my previous minor comment 2, please explain abbreviations also in figure legends. (Especially, in Fig1 and FigS1 e.g. RE)

Point-by-point response to final reviewer comments

REVIEWERS' COMMENTS:

Reviewer #1 (Remarks to the Author):

Designer aptazymes could be useful for a wide range of applications, but engineering novel aptazymes has proven to be a challenge. The authors describe a method for optimizing designs of aptazyme communication modules in which analyte binding to an aptamer module drives a conformation change in the self-cleaving ribozyme domain that either activates or suppresses self-cleavage, leading to mRNAs degradation. The design can handle approximately 200k variants. This is a good number for evaluating relatively small libraries (5-7 randomized positions).

The work is well designed and carefully implemented and provides some new insights. It builds upon prior work in *E. coli* and *S. cerevisiae* in this groups and ligand-gated Twister allostery in mammalian cells in other groups. It is a nice refinement on existing methodologies and an overall positive contribution that should be published.

We thank the reviewer for acknowledging the value of our work and recommending publication of the manuscript.

Reviewer #3 (Remarks to the Author):

The authors answered the most of my previous concerning points. However, there are still a few points that I should indicate.

Major comments

1. Regarding my previous major comment 2, the authors now show that low-ranking Tet-Twister ribozymes have lower performance than high-ranking Tet-Twister ribozymes in new Supplemental Fig. 7. However, in other switches (hammerhead ribozymes, HDV ribozymes, and Gua-U1-snRNP riboswitches), the superiority of high-ranking constructs over low-ranking constructs was not confirmed. The data of low-ranking constructs in other switches will make the study more persuasive.

We thank the reviewer for this suggestion. As stated by the reviewer, Suppl. Fig. 7 clearly shows that constructs that do not survive the positive selection criteria for hit selection, did not show any switch activity, as expected. In contrast, 28 out of 29 constructs selected from the new Gua-HDV, Gua-HHR and Gua-U1snRNP library screening runs clearly demonstrated activity in functional assays. Moreover, our screens successfully recovered previously known functional constructs and demonstrated an overall good correlation between amplicon seq- and cell culture-derived functional data. We therefore feel that the present data convincingly demonstrates our method's capability to identify functional constructs and deselect non-functional ones. We therefore kindly ask the reviewer to refrain from the demand for additional functional characterization work.

Minor comments

1. Regarding my previous major comment 8, the vertical axis of the previous figure 4a (new figure 5b) was properly corrected. However, there is still the same mistake in the vertical axis of the new figure 6d.

We apologize for this omission. The axis labeling in Fig. 6d has now been corrected.

2. While the variant names of Gua-responsive riboswitches in figure 6e-g are described as "GuaU1B1" and "GuaU1B10", in page 18 line 379-380, these variants are described as "GuaB1U1" and "GuaB10U1", respectively.

We thank the reviewer for this hint. The text has now been corrected to match the names in the Figure.

3. Related to my previous minor comment 2, please explain abbreviations also in figure legends. (Especially, in Fig1 and FigS1 e.g. RE)

We apologize for this omission. We have again checked the Figures for abbreviations and have now included respective explanations in the corresponding Figure legends, both, in the main text and the Supplementary information.